# Evaluation of the Available Energy Value and Amino Acid Digestibility of Brown Rice Stored for 6 Years and Its Application in Pig Diets

**DOI:** 10.3390/ani13213381

**Published:** 2023-10-31

**Authors:** Beibei He, Jingjing Shi, Kuanbo Liu, Junlin Cheng, Weiwei Wang, Yongwei Wang, Aike Li

**Affiliations:** Institute of Grain Quality and Nutrition, Academy of National Food and Strategic Reserves Administration, Beijing 100037, China; hbb@ags.ac.cn (B.H.); sjj@ags.ac.cn (J.S.); lkb@ags.ac.cn (K.L.); cjl@ags.ac.cn (J.C.); www@ags.ac.cn (W.W.)

**Keywords:** stored brown rice, energy and nutrient value, growth performance, physiological and metabolic characteristic, meat quality, pig

## Abstract

**Simple Summary:**

The present study evaluated the available energy value and amino acid digestibility of brown rice stored for 6 years and investigated its application in weaned piglets and fully grown pig diets. The results showed that the available energy and ileal digestibility of amino acids did not change in brown rice stored under good conditions for 6 years. Stored brown rice also had no significant effects on growth performance, nutrient-apparent total tract digestibility, serum biochemical indices, carcass traits, meat quality, and muscular fatty-acid profiles of pigs but did reduce the activity of digestive enzymes in the pigs’ small intestines. Altogether, brown rice stored under good conditions for 6 years can be used as a high-quality energy-feed raw material in pig diets.

**Abstract:**

Long-term storage may reduce the nutritional quality of brown rice, so the present study aimed to evaluate the nutritional values of long-term-stored nutrition in pig diets. In Exp. 1, 18 Landrace × Yorkshire (L × Y) barrows with an initial body weight (IBW) of 25.48 ± 3.21 kg were randomly assigned to three treatments, including a corn-based diet, one-year-stored brown rice (BR1) diet, and six-year-stored brown rice (BR6) diet, to determine the digestible energy (DE) and metabolizable energy (ME) values of stored brown rice. In Exp. 2, 24 barrows (L × Y; IBW: 22.16 ± 2.42 kg) fixed with ileal T-cannula were randomly allotted to four dietary treatments, including a corn diet, two stored brown rice diets, and a nitrogen-free diet, to evaluate the amino acid (AA) digestibility of the stored brown rice. In Exp. 3 and 4, 108 crossbred weaned piglets (L × Y; IBW: 9.16 ± 0.89 kg) and 90 crossbred growing pigs (L × Y; IBW: 48.28 ± 3.51 kg) were allotted to three treatment diets, including a control diet and two stored brown rice diets, respectively, to investigate the application of stored brown rice in weaned piglets and fully grown pig diets. The results indicated that there was no significant difference in the DE and ME values between corn and stored brown rice (*p* > 0.05), while the apparent ileal digestibility (AID) of arginine, histidine, asparagine + aspartic acid (Asx), and the standardized ileal digestibility (SID) of arginine and histidine were higher in the stored brown rice diet compared to the corn diet (*p* < 0.05). Compared to the corn, the stored brown rice showed no significant effects on growth performance, nutrient-apparent total tract digestibility (ATTD), and serum biochemical indices (*p* > 0.05) but showed decreased activity in the various digestive enzymes in the duodenum, jejunum, and ileum of the weaned piglets (*p* < 0.05). Also, the stored brown rice diet showed no significant effects on growth performance, carcass traits, meat quality, as well as the fatty acid profiles in the longissimus dorsi muscle of fully grown pigs compared with the corn diet (*p* > 0.05). In conclusion, the brown rice stored for 6 years under good conditions had no obvious changes in the available energy and nutrient values. Although it may reduce digestive enzyme activity in the small intestines of the piglets, the stored brown rice showed no obvious adverse effects on growth performance and meat quality and can be effectively used in pig diets.

## 1. Introduction

With the rapid development of animal husbandry in recent years, the contradiction between the supply and demand of corn has become increasingly prominent [1]. According to data from the Chinese National Statistics Bureauso, in 2022, China’s corn production reached 277 million tons, with a demand of 298 million tons and an import volume of 20.62 million tons. Rice is the most important grain crop, and the annual output has reached more than 200 million tons in China [2]. The inventory of rice in China has remained stable at over 100 million tons in recent years, with an estimation of over 14 million tons of long-term stored rice. On the condition that food rations are absolutely safe, the overstocked grain can be considered as an alternative energy source. Therefore, research on the nutritional characteristics and utilization values of stored rice in animal feeding has great theoretical and practical significance.

Rice is a variety of grain that is not resistant to long-term storage, as aging and deterioration can occur after the second year, and the warranty storage life is about three years. During grain storage, the composition and structure of the main nutrients (protein, starch, and fat) will change due to their own respiration, oxidation, and the action of microorganisms, causing a decrease in nutritional values [3]. Higher disulfide bond content and surface hydrophobicity and lower free sulfhydryl content were produced during storage, which led to the deterioration of the rice’s protein value [4]. Compared to fresh rice, the solubility of total starch and amylose decreased, resulting in a lower gelatinization temperature, greater hardness, and lower viscosity of the rice [5]. Moreover, the lipid oxidation increased the contents of free fatty acids and volatile carbonyl compounds, such as glutaraldehyde and hexanal, which reduced the palatability of the stored grain [6].

Several studies have reported changes in the energy values and nutrient digestibility of stored corn and wheat in animal diets. Zhang et al. reported that the nutrient availability of corn, including the digestible energy (DE) and metabolizable energy (ME) values, decreased after being stored at room temperature for 10 months [7]. Although there were no significant changes in the ME, the digestibility of crude protein, histidine, arginine, and starch decreased quadratically with corn stored for 4 years at room temperature [8]. Some studies have reported that the nutritional value of stored corn and wheat had little change under standard storage conditions [9,10,11]. The major nutrients of brown rice obtained from rice hulling were higher than or equal to corn and can be effectively used as an energy ingredient [12,13]. However, few studies have been completed on stored brown rice regarding the available energy values, amino acid digestibility, and its application in animal diets.

The objective of the present study was to evaluate the available energy value and amino acid (AA) digestibility of brown rice stored for 1 or 6 years and to investigate the effects of stored brown rice on growth performance, nutrient digestibility, serum biochemical parameters, intestinal enzyme activities in weaned piglets, carcass characteristics, and meat quality in fully grown pigs. The hypothesis was that if brown rice was stored under proper conditions, its available energy value and amino acid digestibility would not change significantly, and the growth performance and meat quality of the pigs would not be influenced by the inclusion of stored brown rice in the diets.

## 2. Materials and Methods

All experiments were conducted in accordance with the Chinese Guidelines for Animal Welfare and Experimental Protocol, and prior approval was obtained from the Animal Care and Use Committee of Academy of National Food and Strategic Reserves Administration (ethical approval code: 20230316006).

The paddy rice used in this study was stored in brick concrete barns (36 × 24 × 6 m, storage capacity 5000 tons) at National Grain Reserve Barn of Heilongjiang Province, as it has the greatest rice yield and inventory in China. Rice stored up to 6 years has been selected as a representative of long-term storage rice, and rice stored for 1 year has also been collected as fresh rice. The temperature and relative humidity in all grain barns was controlled within 20 °C and 70% all year around. Before the start of the experiment, the paddy rice was hulled into brown rice and crushed for subsequent experiments. All the diet, feces, and digesta samples collected in animal experiments were ground to pass through a 1 mm sieve for chemical analysis (Table A1). Dry matter (DM, method 934.01), crude protein (CP, method 990.03), ash (method 942.05), ether extract (EE, method 920.39), calcium (method 985.01), and total phosphorus (method 985.01) contents were determined according to the procedures of Association of Official Analytical Chemists (AOAC) International (2006) [14]. The contents of 18 AAs were measured according to the methods in AOAC. Tryptophan was hydrolyzed with LiOH at 110 °C for 22 h and then analyzed using High-Performance Liquid Chromatography (Agilent 1200 Series, Santa Clara, CA, USA). Cysteine and methionine were firstly oxidized with performic acid and hydrolyzed with 7.5 mol/L HCl at 110 °C for 24 h and then analyzed using AA analyzer. Another 15 AAs were hydrolyzed with 6 mol/L HCl at 110 °C for 24 h firstly and then analyzed using AA analyzer (L-8900, Hitachi; Tokyo, Japan). Neutral detergent fiber (NDF) was determined using α-amylase treated method without correction for insoluble ash, and acid detergent fiber (ADF) was expressed as inclusive of residual ash according to the procedures of the ANKOM200 Fibre Analyzer (Ankom Technology, Macedon, NY, USA). The gross energy (GE) was analyzed using automatic isoperibol oxygen bomb calorimeter (IKA C6000; IKA, GER). The fatty acid value was analyzed according to the procedures of GB/T 20569-2006 (Guidelines for evaluation of paddy storage character) [15].

### 2.1. Exp. 1: Evaluating the DE and ME Values of Stored Brown Rice

Eighteen Landrace × Yorkshire (L × Y) barrows with an initial body weight (IBW) of 25.48 ± 3.21 kg were randomly allotted to 3 treatment groups with 6 replicated pigs per treatment. The diet was formulated to contain 96.9% corn (Corn) or brown rice stored for 1 (BR1) or 6 years (BR6) and 3.1% minerals and vitamins to meet the nutrient requirements for growing pigs recommended by the National Research Council (NRC, 2012) (Table A2) [16]. Each pig was individually raised in metabolism crates (1.4 × 0.45 × 0.6 m), and the room temperature was controlled at 23 ± 2 °C. Pigs were provided the feed equivalent of 4% of their IBW daily and fed twice at 08:30 and 16:30, respectively.

Pigs were first fed commercial diet for 7 d to adapt to the metabolic chamber. Animal experiments lasted for 12 d, of which 7 d were used to adapt to the experimental diet and 5 d to collect feces and urine samples [17]. Feces sample of each pig was collected separately and stored at −20 °C. Urine of each pig was collected in barrels containing 50 mL 6 mol/L HCl, and 10% of the total urine collected daily was stored at −20 °C. At the end of the experiment, the fecal and urine samples were thawed and merged to obtain the sub-sample by pig. Fecal sub-sample was dried at 65 °C for 3 d using a drying oven while 4 mL of urine sample was added onto a quantitative filter paper in crucibles and then dried at 65 °C for 8 h using a drying oven for analysis of GE.

### 2.2. Exp. 2: Evaluating the AA Digestibility of Stored Brown Rice

Eighteen barrows (L × Y) with T-cannula at the terminal ileum (IBW: 22.16 ± 2.42 kg) were allocated to 3 experiment diets in a completely randomized design with 6 replicated pigs per treatment. The diets were formulated to contain 96.6% of corn (Corn) or brown rice stored for 1 (BR1) or 6 years (BR6) and 3.1% minerals and vitamins to meet the nutritional requirements for growing pigs recommended by the NRC (2012) [16]. N-free diet containing 73.35% corn starch and 15% sucrose was used to evaluate the losses of basal ileal endogenous N and AAs. In addition, 0.3% chromic oxide (Cr_2_O_3_, ≥99.0%; 10,006,918, SINOPHARM, Beijing, China) was included as exogenous indicator (Table A2). Feeding management was the same as above.

After a 15 d recovery postoperative period, pigs were fed commercial diet for 7 d to adapt to the environment. The animal experiment lasted for 7 d, of which 5 d were used to adapt to the experimental diets and 2 d to collection of digesta, which lasted for 9 h daily beginning at 08:30 [17]. The sample bag was fixed to the cannula to collect the digesta and then stored at −20 °C. After the collection period, the digesta samples were thawed and merged to obtain the sub-sample by pig, and then lyophilized by vacuum freeze drier. The contents of chromium in the diets and digesta samples were measured using an Atomic Absorption Spectrophotometer (model Z-5000, Hitachi Corp., Tokyo, Japan) according to the method of García-Rico et al. [18]

### 2.3. Exp. 3: Growth Trail on Weaned Piglets

One hundred and eight weaned piglets (L × Y, male; IBW: 9.16 ± 0.89 kg) were selected from a commercial herd, and randomly allocated into 3 diet treatments with 6 replicate pens per treatment and 6 pigs per pen (2.1 × 1.8 × 0.6 m, 0.5 m above the ground). The treatment included 1 control diet (Control) and 2 experimental diets formulated by completely replacing corn with brown rice stored for 1 (BR1) or 6 years (BR6), respectively (Table 1). The diets were formulated based on the ME value in Exp. 1 and the SID AA value in Exp. 2, and meet the nutritional requirements for weaned piglets recommended by NRC (2012) [16]. The ME and SID lysine, methionine, threonine, and tryptophan in 3 diets were kept the same. In the last 2 weeks of the experiment, 0.3% chromic oxide was added to each diet as an exogenous indicator.

Pigs were housed in pens with drinkers, feeders, and slatted floors, and were provided water and feed freely. The environment temperature was controlled at 22 ± 2 °C. The experiment lasted for 28 days. Pigs and feed were weighed at the beginning (d 1) and the end of the experiment (d 28) to calculate average daily gain (ADG), average daily feed intake (ADFI), and feed conversion ratio (FCR).

From d 25 to d 27, approximately 100 g of fresh feces were collected daily from each pen and immediately stored at −20 °C. All samples were pooled by pen and then dried at 65 °C in a drying oven for 72 h. After fasting for 16 h, blood samples were collected through intravenous puncture on the morning (07:00) of d 28 and then injected into a 10 mL vacuum tube. After centrifugation at 3000× *g* for 15 min (4 °C), serum samples were collected and stored at −20 °C for the further determination of biochemical parameters. At the end of the experiment, 18 pigs with nearly average BW were selected from each pen. After overnight fasting, the pigs were slaughtered, and the gastrointestinal tract was ligated; then, the mucosa of duodenum, jejunal, and ileum were scraped with a glass slide and stored in liquid nitrogen.

After fasting overnight, the pigs are slaughtered, and the gastrointestinal tract of each pig is ligated. Then, the mucosa of the duodenum, jejunum, and ileum is scraped with a slide and stored in liquid nitrogen.

### 2.4. Exp. 4: Growth Trail on Fully Grown Pigs

Ninety growing pigs (L × Y; IBW: 48.28 ± 3.51 kg) were randomly allotted into 3 dietary treatments with 6 replicate pens per treatment and 5 pigs per pen (4.0 × 2.8 × 1 m). The treatment diets included 1 control diet (Control) and 2 experimental diets formulated by completely replacing corn with brown rice stored for 1 (BR1) or 6 years (BR1), respectively (Table 2). The diets were formulated based on the ME value in Exp. 1 and the SID AAs value in Exp. 2 and meet the nutritional requirements for pigs in different phases recommended by NRC (2012) [16]. The ME and SID lysine, methionine, threonine, and tryptophan in 3 diets were kept the same.

Pigs were kept in pens with drinkers, feeders, and slatted floors and were provided water and feed freely. The environment temperature was controlled at 18 ± 2 °C. The experiment lasted for 8 weeks and included 2 phases: I (d 1 to d 24); II (d 25 to d 56). At the beginning (d 1) and end of each phase (d 24 and d 56), pigs and feed were weighed to determine ADG, ADFI, and FCR.

At the end of the experiment, 15 pigs with near-average BW were selected from each pen. After overnight fasting, pigs were subjected to electric shock (250 V, 0.5 A, for 5–6 s), bleeding, and evisceration using standard commercial procedure. Approximately 10 g of longissimus dorsi muscle (LDM) were sampled from the left half of each carcass and stored at −20 °C.

### 2.5. Determination of Serum Biochemical Indices and Intestinal Enzyme Activities

Serum albumin (ALB), globulin (GLB), total protein (TP), triglyceride (TG), total cholesterol (TC), alanine aminotransferase (ALT), aspartate aminotransferase (AST) and urea nitrogen (UN) were measured by automatic biochemical analyzer (7020 series; Hitachi, Japan) and following the protocol of assay kits purchased from Nanjing Jiancheng Bioengineering Institute (Nanjing, China). The content of serum immunoglobulin (Ig) A, IgG, and IgM were determined using enzyme-linked immunosorbent assay (ELISA) using assay kits purchased from Takara Biomedical Technology Institute (Beijing, China). Serum catalase (CAT), total antioxidant capacity (T-AOC), glutathione (GSH), glutathione peroxidase (GSH-PX), malondialdehyde (MDA), superoxide dismutase (SOD) were measured using assay kits purchased from Nanjing Jiancheng Bioengineering Institute (Nanjing, China).

The duodenum, jejunal, and ileum mucosa samples were homogenized in cold maleic acid buffer (0.1 mol/L, pH = 6.8, 1: 10, *w*/*v*) and then centrifuged at 3000× *g* for 10 min. Supernatants were collected to evaluate the activities of amylase, lipase, chymotrypsin, trypsin, lactase, maltase, and sucrase, following the protocol of assay kits purchased from Nanjing Jiancheng Bioengineering Institute (Nanjing, China).

### 2.6. Determination of Carcass Characteristics and Meat Quality

After slaughter and scraping, the head, hooves, tail, and internal organs of pigs were removed, while the suet and kidneys were preserved to record carcass weight, and dressing percentage was calculated by dividing carcass weight by live body weight. Carcass straight length was measured from the first rib to the end of the public bone. Backfat thickness was recorded at the first rib, last rib, and last lumbar vertebra, and Loin eye height and width were measured at the 10th rib following the equation loin–eye area (cm^2^) = 0.7 × loin eye height (cm) × loin eye width (cm), according to the NY/T 825-2004 (Technical Regulation for Testing of Carcass Traits in Lean-Type Pig) [19].

The LDM on the left half of carcass between the 10th and 12th ribs were sampled for analysis of meat quality, including drip loss, shear force, pH, and muscle color, according to the NY/T 821-2019 (Technical Code of Practice for Pork Quality Assessment) and NY/T 1180-2006 (Determination of Meat Tenderness Shear Force Method) [20,21]. About 30 g of meat was hung in a plastic bag at 4 °C for 24 h and kept out of contact with the bag. Drip loss was calculated as a percentage of the droplet amount compared to the initial meat weight. Meat was cooked in a water bath at 70 °C for 20 min, and then ten cylindrical samples were obtained by cutting the meat parallel to the fiber direction, and shear force was determined by cutting the cylindrical sample vertically to the myofiber axis using a digital-display-muscle tenderness meter. At 45 min postmortem, initial pH of LDM was recorded using a glass penetration pH electrode, and pH of LDM was detected again at 24 h postmortem. The meat color was measured as L* (lightness), a* (redness), and b* (yellowness) using a tristimulus colorimeter three times at 24 h postmortem. About 20 g meat sample was lyophilized to determine the fatty acids profile using classical gas chromatography (6890 series; Agilent Technologies, Wilmington, DE, USA) [22].

### 2.7. Statistical Analysis

PROC UNIVERSATE program (SAS Inst. Inc., Carry, NC, USA) of SAS 9.2 was used to check the normal and abnormal values of growth performance, nutrient digestibility, serum biochemical index, enzyme activity, carcass traits, and meat quality data. Cook’s distance and abandonment method was used to identify outliers. Then, the PROC GLM program of SAS was used to analyze the data. The diet was the only fixed effect, while each pig was considered an experimental unit (for growth performance data, each pen was considered an experimental unit). The LSMEANS statement was used to separate treatment means, and the Tukey test was used to adjust the data. Significant difference was declared at *p* < 0.05.

## 3. Results

The chemical compositions of the corn and stored brown rice are shown in Table A1. The contents of the DM, ash, calcium, and total phosphorus of the stored brown rice had no differences compared with the corn. The contents of CP and GE were higher, while the EE, NDF, and ADF contents were lower in the stored brown rice compared to the corn. The contents of lysine, tryptophan, and arginine were higher, while the contents of leucine, phenylalanine, glutamine + glutamic acid (Glx), and proline were lower in the stored brown rice compared to the corn. Moreover, all the chemical compositions showed no obvious differences between the brown rice stored for 1 or 6 years, except for the fatty acid values of the brown rice stored for 6 years, which was higher compared to the brown rice stored for 1 year (26.43 vs. 19.73 mg KOH/100 g) (Table A1).

### 3.1. Available Energy Value and AA Digestibility of Stored Brown Rice

The DE values of the brown rice stored for 1 or 6 years were 14.70 and 14.88 MJ/kg (as-fed basis), while the ME values of the brown rice stored for 1 or 6 years were 14.22 and 14.31 MJ/kg, respectively. The DE and ME values of the corn were 14.52 and 14.17 MJ/kg, which are slightly lower than that of the stored brown rice (Table 3). The AID and SID values of arginine and histidine and the AID values of asparagine + aspartic acid in the stored brown rice were higher than those of the corn (*p* < 0.05). The digestibility of all the amino acids in the brown rice showed no significant differences between the brown rice stored for 1 or 6 years (*p* > 0.05) (Table 4).

### 3.2. Growth Performance and Apparent Total Tract Digestibility (ATTD) of Nutrients of Weaned Piglets Fed Stored Brown Rice Diets

Compared with the corn, the stored brown rice showed no significant influences on the final BW, ADG, ADFI, and FCR of the weaned piglets (*p* > 0.05), and there were also no significant differences in the growth performance between piglets fed with brown rice stored for 1 or 6 years (*p* > 0.05) (Table 5). There were no differences in the ATTD of the DM, GE, CP, OM, NDF, ADF, Ca, and *p* between piglets fed corn or stored brown rice diets (*p* > 0.05) (Table 6).

### 3.3. Serum Biochemical Parameters

As shown in Table 7, the concentrations of TP, ALB, GLB, TG, TC, AST, ALT, and UREA in the serum of the piglets showed no significant differences between the corn and stored brown rice groups (*p* > 0.05), and the activities of CAT, TAOC, GSH, GSH-PX, MDA, and SOD were also not different between the piglets fed the stored brown rice diets or the control diet (*p* > 0.05). However, the concentration of serum IgG decreased in weaned piglets fed stored brown rice diets compared to the corn diet group (*p* < 0.01).

### 3.4. Digestive Enzymes Activities

As shown in Table 8, compared to the piglets fed corn diets, the activities of lipase, chymotrypsin, lactase, maltase, and sucrase in the duodenum, as well as the activities of lactase and maltase in the jejunum, decreased in the piglets fed two stored brown rice diets (*p* < 0.05). What is more, compared to the piglets fed corn diets, the activities of trypsin and sucrase, as well as the activities of lactase and maltase in the ileum, decreased in the jejunum of piglets fed the 6-year-stored brown rice diets (*p* < 0.05). For the two stored brown rice diets, the activity of lactase in the duodenum, as well as the activity of lactase in the ileum of piglets fed 6-year-stored brown rice, was lower compared to the piglets fed 1-year-stored brown rice (*p* < 0.05).

### 3.5. Growth Performance, Carcass Traits, and Meat Quality of Fully Grown Pigs Fed Stored Brown Rice Diets

Compared to pigs fed two stored brown rice diets, the final BW and ADFI in phase I and ADFI in the total phase were markedly decreased in pigs fed a corn diet (*p* < 0.05). The BW, ADG, ADFI, and FCR of fully grown pigs in all phases showed no significant differences between pigs fed brown rice stored for 1 or 6 years (*p* > 0.05) (Table 9).

The dressing percentage of LDM tended to decrease in pigs fed a corn diet compared with those fed the two stored brown rice diets (*p* = 0.06), while other carcass traits, including the carcass straight length, backfat thickness, and loin–eye area, were not influenced by the corn or stored brown rice diets (*p* > 0.05) (Table 10). The meat quality, including the drip loss, shear force, pH, and meat color, was also not influenced by different diet treatments (*p* > 0.05) (Table 10). The corn diet and the two stored brown rice diets also had no significant effect on the fatty acid concentrations in the LDM of fully grown pigs (*p* > 0.05) (Table 11).

## 4. Discussion

China has a large inventory of stored rice, which has great potential for replacing corn as an energy feed. However, there is a lack of data on the nutritional value and feeding performance of long-term stored rice. The present research proves that brown rice stored for 6 years under good conditions has no obvious changes in available energy and nutrient values and shows no adverse effects on growth performance and meat quality in pigs, which provides a theoretical basis for the efficient use of stored rice in livestock production.

### 4.1. Available Energy Values and Amino Acid Digestibility of Corn and Stored Brown Rice

In the present study, the contents of CP, GE, total phosphorus, and several essential amino acids, as well as the digestibility of arginine, histidine, and aspartic acid in stored brown rice, were higher than that of the corn (*p* < 0.05) (Table A1 and Table 4), which was consistent with previous studies and indicates that brown rice is an alternative energy feed to corn [23,24,25]. There were no significant differences in the main nutrient contents between the two stored brown rice diets (Table A1), except for an increase in the fatty acid value in brown rice stored for 6 years, which means that long-term storage did not change the total amount of protein, starch, and fat but may increase lipid oxidation in brown rice [26]. Bartov reported that the storage of maize for up to 110 months under good conditions had no adverse effects on the main nutrient contents and the ME value for young male broiler chicks [9]. Mitchell and Beadles showed that the nutrient value of wheat and corn had no obvious changes over long periods of time when being stored under conditions that prevented insect infestation and mold growth [11]. Yin et al. also showed that the digestibility of starch, CP, and the apparent metabolic energy value of corn in chicken was also not affected by storage for 5 years [8]. These research analyses are consistent with the present results that long-term storage under proper conditions might not cause a decrease in available energy and nutrient digestibility in stored brown rice (Table 3 and Table 4).

### 4.2. Effects of Stored Brown Rice on Growth Performance and Nutrient Digestibility of Piglets

When replacing 100% of the corn with stored brown rice, the growth performance and nutrient digestibility of the weaned piglets showed no significant difference between the dietary treatments (*p* > 0.05) (Table 7 and Table 8). Kim et al. replaced 50% of the corn with brown rice and found that the final BW, overall ADG, and ATTD of the DM and GE were higher in the weaned pigs fed a brown rice diet [27]. Sittiya et al. showed that brown rice could totally replace corn in chicken diets without a negative effect on growth performance [13]. Rice and broken rice can also be efficiently used in livestock and poultry diets [28,29]. Although brown rice is rich in various phytochemicals, such as anthocyanins and tocopherols [30], it can be less palatable compared to corn, which may cause a decrease in the FI of weaned piglets [31]. At the same time, some studies have shown that the extrusion and heat treatment of brown rice had a significant effect on its nutrient utilization, which should be paid attention to when brown rice is used in animal feeds [32,33]. However, long-term storage at an ambient temperature may induce a decrease in the apparent metabolizable energy of wheat-distiller-dried grains with solubles; however, there were no serious effects on the feeding value in broiler chickens [34], which indicates that the growth performance of animals may be poorly correlated to the changes in the nutrient values of stored grains [35,36]. When the corn was stored for 4 years, it had no significant effect on the FI, body weight gain, and FCR of broilers from 0 to 6 weeks [8], which is consistent with the present results (Table 5) that the brown rice stored for 6 years under proper conditions had no serious effects on the growth performance of weaned piglets.

### 4.3. Effects of Stored Brown Rice on Serum Profiles and Intestinal Enzyme Activities of Piglets

There were no significant changes in the serum profiles and antioxidant indexes between the piglets fed the stored brown rice diets relative to the control diet (*p* > 0.05) (Table 9), which is consistent with previous research analyses [27,37]. Several research analyses have shown that the concentration of plasma glucose and insulin, as well as the activities of maltase and aminopeptidase in the jejunum of rice-fed piglets, were higher than that of corn-fed piglets [38,39,40], which may be due to a higher villus height and result in a higher ADG of the piglets. Different from these previous studies, the present results showed that most of the activities of lipase, chymotrypsin, lactase, maltase, and sucrase decreased in the small intestines of piglets fed the stored brown rice diets compared to the corn diet (*p* < 0.05) (Table 10), which is possibly due the different crushing particle size of brown rice, as the particle size had a significant effect on the nutrient utilization of the brown rice when it was used in the animal feed [41,42]. What is more, the activities of lactase in the duodenum and lactase in the ileum of the piglets fed the brown rice stored for 6 years was lower compared to the brown rice stored for 1 year (*p* < 0.05) (Table 10). Although there have been no related research analyses on the effects of brown rice on intestinal enzyme activities in pigs, a previous study has shown that the fewer non-starch polysaccharides and resistant starch in brown rice make it easier to digest compared to corn [43], so the decrease in mucosal enzyme activities caused by brown rice feeding may not lead to changes in nutrient digestibility and growth performance of the piglets. He et al. found that extrusion or enzyme supplementation in a stored brown rice diet significantly increased the carbohydrase activity in the digestive tract of piglets [44]. Dadalt et al. found that diets supplemented with multicarbohydrase and phytase improved the digestibility of energy and some nutrients of broken rice in post-weaned piglets [45]. Therefore, it is possible to increase the nutrient digestibility of stored brown rice by appropriately extruding or supplementing enzyme preparations when used in feed.

### 4.4. Effects of Stored Brown Rice on Growth Performance, Carcass Traits, and Meat Quality of Fully Grown Pigs

In the present study, the growth performance of fully grown pigs in all phases showed no differences between pigs fed brown rice stored for 1 or 6 years (*p* > 0.05) (Table 11). Kim et al. replaced corn with 50%, 75%, and 100% brown rice, respectively, and found that there were no differences in growth performance, nutrient digestibility, and carcass characteristics among dietary treatments [27]. However, the long-term feed of brown rice may have an effect on the intestinal microbiota of pigs since the low fiber contents in brown rice compared to corn provide less substrates for bacterial fermentation in the hindgut. Yin et al. showed that the catalase activity and peroxidase activity decreased, and the acidity of fatty acids increased in corn stored for 5 years, which may result in a significantly lower pH and increased drip loss in broiler breast muscles [8]. However, in the present study, although the fatty acid value increased significantly in brown rice stored for 6 years compared to brown rice stored for 1 year (Table A1), the pigs fed the brown rice stored for 1 or 6 years showed no significant differences in carcass traits, meat quality, and fatty acid concentrations in the LDM (*p* > 0.05) (Table 10 and Table 11).

The present results showed that there were no significant changes in the nutritional values and utilization performance of stored brown rice in pig feeds; however, there are still some problems that need to be solved. For example, due to differences in the origin, quality, and storage year of the stored rice purchased by feed and breeding enterprises, as well as different crushing processes of each enterprise, the sampling range and representativeness should be further expanded and studied to comprehensively and systematically reveal the changes in nutritional values and feeding quality of stored grains in China.

## 5. Conclusions

The brown rice stored under good conditions for 6 years revealed no obvious changes in available energy values and amino acid digestibility; furthermore, although it might reduce the digestive enzyme activities in the small intestine of weaned piglets, it had no significant effects on growth performance, ATTD, and the serum biochemical indices of weaned piglets and showed no adverse effects on the carcass traits, meat quality, and muscle fatty acid profiles of fully grown pigs. Therefore, brown rice stored under good conditions can be used as an alternative energy feed in pig diets.

## Figures and Tables

**Table 1 animals-13-03381-t001:** Ingredients and analyzed nutrient levels of the experimental diets used in Exp. 3 (%, as-fed basis) ^1^.

Item	Control	BR1	BR6
Ingredients			
Corn	62.78	-	-
Soybean meal (46% CP)	15.90	15.90	15.90
Extruded soybean	9.00	8.40	7.72
BR1	-	63.85	-
BR6	-	-	65.02
Fish meal	4.00	4.00	4.00
Whey powder	4.00	4.00	4.00
Soybean oil	1.00	0.60	0.60
Dicalcium phosphate	0.76	0.66	0.66
Limestone	0.68	0.78	0.78
Sodium chloride	0.30	0.30	0.30
Chromic oxide	0.30	0.30	0.30
Vitamin and mineral permix ^2^	0.50	0.50	0.50
L-Lysine HCl	0.46	0.42	0.44
DL-Methionine	0.07	0.06	0.08
L-Threonine	0.21	0.21	0.24
L-Tryptophan	0.04	0.02	0.03
Analyzed nutrient levels			
DM	88.85	89.75	89.47
CP	18.82	18.66	18.70
GE, MJ/kg	16.35	16.52	16.43
Calcium	0.70	0.68	0.64
Total phosphorus	0.62	0.63	0.63
Calculated nutrient levels ^3^			
ME, MJ/kg	13.72	13.73	13.71
SID Lysine	1.23	1.23	1.23
SID Methionine	0.36	0.36	0.36
SID Threonine	0.74	0.73	0.73
SID Tryptophan	0.21	0.21	0.21

^1^ BR1 = Brown rice stored for 1 year; BR6 = Brown rice stored for 6 years; DM = dry matter; CP = crude protein; GE = gross energy; ME = Metabolizable energy; SID = Standardized ileal digestible. ^2^ Vitamin and mineral premix provided the following per kg of diet: vitamin A, 12,000 IU; vitamin D3, 2000 IU; vitamin E, 24 IU; vitamin K3, 2 mg; vitamin B12, 24 μg; vitamin B2, 6 mg; vitamin B1, 2 mg; vitamin B5, 20 mg; vitamin B6, 3 mg; niacin acid, 30 mg; choline chloride, 0.4 mg; folic acid, 3.6 mg; biotin, 0.1 mg; Mn, 40 mg (as manganese oxide); Fe, 96 mg (as ferrous sulfate); Zn, 120 mg (as zinc oxide); Cu, 8 mg (as copper sulfate); I, 0.56 mg (as ethylenediamine dihydroiodide) and Se, 0.4 mg (as sodium selenite). ^3^ These values were calculated from data provided in Exp. 1 and 2.

**Table 2 animals-13-03381-t002:** Ingredients and analyzed nutrient levels of the experimental diets used in Exp. 4 (%, as-fed basis) ^1^.

Item	Growing Phase: 50 to 75 kg			Growing Phase: 75 to 100 kg		
	Control	BR1	BR6	Control	BR1	BR6
Ingredients						
Corn	68.41	-	-	75.60	-	-
Soybean meal	21.00	21.00	21.00	16.68	16.05	16.00
Extruded soybean	3.10	2.30	1.60	-	-	-
Wheat bran	4.00	4.00	4.00	4.00	4.00	4.00
BR1	-	70.54	-	-	77.85	-
BR6	-	-	70.32	-	-	81.85
Soybean oil	1.20	-	-	1.58	0.10	-
Dicalcium phosphate	0.83	0.78	0.79	0.62	0.50	0.56
Limestone	0.80	0.82	0.82	0.80	0.88	0.86
Sodium chloride	0.35	0.35	0.35	0.35	0.35	0.35
Vitamin and mineral ^2^	0.50	0.50	0.50	0.50	0.50	0.50
L-Lysine HCl	0.19	0.14	0.17	0.23	0.18	0.20
DL-Methionine	0.02	0.00	0.03	0.02	-	0.03
L-Threonine	0.02	0.02	0.06	0.04	0.04	0.08
L-Tryptophan	0.03	0.00	0.01	0.03	-	0.02
Analyzed nutrient levels						
DM	86.35	87.52	87.04	88.30	87.21	88.29
CP	16.46	16.40	16.39	14.11	14.08	14.12
GE, MJ/kg	16.13	16.05	16.09	16.03	16.06	16.18
Calcium	0.60	0.59	0.59	0.52	0.50	0.50
Total phosphorus	0.53	0.55	0.56	0.47	0.48	0.48
Calculated nutrient levels ^3^						
ME, MJ/kg	13.25	13.25	13.25	13.10	13.10	13.10
SID Lysine	0.85	0.85	0.85	0.73	0.73	0.73
SID Methionine	0.26	0.26	0.26	0.24	0.24	0.24
SID Threonine	0.52	0.52	0.52	0.46	0.46	0.46
SID Tryptophan	0.19	0.19	0.19	0.16	0.16	0.16

^1^ BR1 = Brown rice stored for 1 year; BR6 = Brown rice stored for 6 years; DM = dry matter; CP = crude protein; GE = gross energy; ME = Metabolizable energy; SID = Standardized ileal digestible. ^2^ Vitamin and mineral premix provided the following per kg of diet: vitamin A, 5512 IU; vitamin D3, 2200 IU; vitamin E, 30 IU; vitamin K3, 2.2 mg; vitamin B12, 27.6 μg; vitamin B2, 27.6 mg; vitamin B1, 1.5 mg; vitamin B5, 14 mg; vitamin B6, 3 mg; niacin acid, 30 mg; choline chloride, 400 mg; folic acid, 0.7 mg; biotin, 44 μg; Mn, 40 mg (as manganese oxide); Fe, 75 mg (as ferrous sulfate); Zn, 75 mg (as zinc oxide); Cu, 100 mg (as copper sulfate); I, 0.3 mg (as potassium iodide) and Se, 0.3 mg (as sodium selenite). ^3^ Nutrient levels were analyzed values, except metabolizable energy values were calculated.

**Table 3 animals-13-03381-t003:** Available energy concentration of stored brown rice (MJ/kg, Exp. 1) ^1^.

Item	Corn	BR1	BR6	SEM	*p* Value
DE	14.52	14.70	14.88	0.14	0.06
ME	14.17	14.22	14.31	0.10	0.59
ME/DE	97.56	96.78	96.14	1.31	0.57

^1^ Values are the means of 6 observations. BR1 = Brown rice stored for 1 year; BR6 = Brown rice stored for 6 years; SEM = standard error of the mean; DE = digestible energy; ME = Metabolizable energy.

**Table 4 animals-13-03381-t004:** Apparent and standardized ileal digestibility of crude protein and amino acids in stored brown rice (%, dry-matter basis, Exp. 2) ^1^.

Item	Apparent Ileal Digestibility					Standardized Ileal Digestibility				
	Corn	BR1	BR6	SEM	*p* Value	Corn	BR1	BR6	SEM	*p* Value
CP	72.17	80.03	81.94	3.98	0.06	79.15	89.71	88.07	5.11	0.12
Lysine	58.21	64.15	65.53	6.70	0.52	67.43	76.41	79.39	3.87	0.06
Methionine	75.98	82.85	77.57	4.66	0.33	79.55	87.08	81.48	3.71	0.26
Threonine	68.97	70.96	69.69	5.45	0.93	77.76	82.67	79.77	5.15	0.49
Trptophan	51.97	52.20	54.55	4.14	0.79	63.32	62.08	65.39	4.39	0.48
Leucine	79.62	76.20	77.03	3.09	0.53	86.21	84.58	83.45	2.56	0.65
Valine	69.01	75.82	77.55	3.97	0.11	77.59	81.97	81.79	2.90	0.51
Phenylalanine	76.37	78.60	78.68	3.12	0.71	84.75	84.32	84.07	2.64	0.98
Isoleucine	69.49	74.36	73.45	4.23	0.49	78.10	80.16	78.99	3.48	0.91
Arginine	70.54 ^b^	78.49 ^a^	78.13 ^a^	3.23	0.04	70.91 ^b^	86.38 ^a^	85.96 ^a^	5.12	<0.01
Histidine	77.70 ^b^	86.98 ^a^	87.91 ^a^	2.53	<0.01	86.20 ^b^	97.31 ^a^	96.07 ^a^	3.51	<0.01
Glx (glutamine + glutamic acid)	79.24	79.69	79.23	2.88	0.98	87.17	85.50	84.94	2.59	0.78
Tyrosine	73.17	84.93	82.05	5.96	0.16	82.44	91.39	88.14	5.13	0.44
Serine	68.01	71.35	70.94	4.30	0.70	78.58	79.02	78.61	3.43	1.00
Glycine	47.98	59.29	56.95	5.62	0.14	54.17	66.60	64.60	7.20	0.24
Proline	62.40	62.39	62.34	8.88	1.00	67.99	70.78	64.60	9.59	0.81
Cysteine	70.17	62.16	67.80	7.78	0.58	80.76	71.12	75.83	12.10	0.57
Alanine	67.23	63.72	64.85	5.16	0.79	77.44	71.48	72.83	4.94	0.59
Asx (asparagine + aspartic acid)	66.80 ^b^	77.03 ^a^	77.07 ^a^	4.83	<0.01	76.15	83.09	83.06	3.15	0.10

^1^ Values are the means of 6 observations. BR1 = Brown rice stored for 1 year; BR = Brown rice stored for 6 years; SEM = standard error of the mean; CP = crude protein. ^a, b^ Different superscript letters mean significantly different (*p* < 0.05).

**Table 5 animals-13-03381-t005:** Effects of stored brown rice on growth performance of the piglets (Exp. 3) ^1^.

Item	Control	BR1	BR6	SEM	*p* Value
BW d 0, kg	9.12	9.33	9.04	0.20	0.15
BW d 28, kg	18.30	19.27	19.64	0.31	0.26
ADG, g/d	328.04	368.29	359.26	35.63	0.39
ADFI, g/d	554.78	609.80	610.36	80.32	0.18
FCR	1.71	1.66	1.70	0.26	0.92

^1^ Values are the means of 6 observations. BR1 = Brown rice stored for 1 year; BR6 = Brown rice stored for 6 years; SEM = standard error of the mean; BW = body weight; ADG = average daily gain; ADFI = average daily feed intake; FCR = feed conversion ratio.

**Table 6 animals-13-03381-t006:** Effects of stored brown rice on nutrient digestibility in the piglets (Exp. 3) ^1^.

Item	Control	BR1	BR6	SEM	*p* Value
DM	86.79	86.98	87.95	0.65	0.19
GE	87.29	86.43	87.94	0.44	0.10
CP	81.40	81.12	81.50	0.47	0.71
OM	87.34	88.79	88.61	0.77	0.16
NDF	62.12	61.15	62.25	1.19	0.61
ADF	59.60	58.73	58.89	1.08	0.70
Calcium	43.57	41.57	41.82	1.78	0.74
Phosphorus	40.44	39.76	39.88	0.90	0.73

^1^ Values are the means of 6 observations. BR1 = Brown rice stored for 1 year; BR6 = Brown rice stored for 6 years; SEM = standard error of the mean; DM = dry matter; GE = gross energy; CP = crude protein; OM = organic matter; NDF = neutral detergent fiber; ADF = acid detergent fiber.

**Table 7 animals-13-03381-t007:** Effects of stored brown rice on serum biochemical indices in the piglets (Exp. 3) ^1^.

Item	Control	BR1	BR6	SEM	*p* Value
Biochemical indices					
TP, g/L	37.63	36.06	36.00	3.61	0.88
ALB, g/L	18.56	16.56	17.73	1.45	0.42
GLB, g/L	19.06	19.50	18.27	2.55	0.89
TG, mmol/L	0.66	0.71	0.60	0.13	0.70
TC, mmol/L	1.35	1.65	1.68	0.24	0.36
AST, U/L	59.05	61.79	44.63	13.62	0.43
ALT, U/L	47.97	42.11	31.97	7.00	0.12
Urea, mmol/L	1.84	2.56	1.96	0.41	0.22
Immunity indices, g/L					
IgA	1.76	1.69	1.69	0.05	0.27
IgG	9.22 ^a^	8.53 ^b^	8.50 ^b^	0.14	<0.01
IgM	0.75	0.72	0.72	0.03	0.53
Antioxidant indices					
CAT, U/ml	64.58	51.57	55.98	5.37	0.10
TAOC, U/mL	10.12	9.36	9.09	0.58	0.23
GSH, μmol/L	9.60	8.68	8.68	0.59	0.26
GSH-PX, U/mL	323.49	294.52	318.20	14.86	0.17
MDA, nmol/mL	3.43	3.89	3.75	0.24	0.21
SOD, U/mL	75.78	65.67	70.10	4.72	0.16

^1^ Values are the means of 6 observations. BR1 = Brown rice stored for 1 year; BR6 = Brown rice stored for 6 years; SEM = standard error of the mean; TP = total protein; ALB = albumin; GLB = globulin; TG = total triglyceride; TC = total cholesterol; AST = aspartate aminotransferase; ALT = alanine aminotransferase; Ig = immunoglobulin; CAT = catalase; TAOC = total antioxidant capacity; GSH = glutathione; GSH-PX = glutathion peroxidase; MDA = malonaldehyde; SOD = superoxide dismutase. ^a, b^ Different superscript letters mean significantly different (*p* < 0.05).

**Table 8 animals-13-03381-t008:** Effects of stored brown rice on digestive enzymes in the piglets (Exp. 3) ^1^.

Item	Control	BR1	BR6	SEM	*p* Value
Duodenum					
Amylase, U/g	79.62	70.95	56.71	28.62	0.73
Lipase, U/mg	15.10 ^a^	13.92 ^ab^	12.19 ^b^	0.86	0.02
Chymotrypsin, U/mg	39.61 ^a^	28.35 ^b^	24.67 ^b^	3.84	<0.01
Trypsin, U/mg	16.89	14.72	13.55	1.27	0.07
Lactase, U/g	499.96 ^a^	334.62 ^b^	286.37 ^c^	7.14	<0.01
Maltase, U/mg	215.53 ^a^	183.13 ^ab^	158.66 ^b^	19.54	0.05
Sucrase, U/mg	229.78 ^a^	191.51 ^b^	164.97 ^b^	12.55	<0.01
Jejunum					
Amylase, U/g	44.03	42.21	40.06	19.99	0.98
Lipase, U/mg	15.53	14.01	12.59	1.32	0.14
Chymotrypsin, U/mg	31.65	26.83	22.88	3.18	0.06
Trypsin, U/mg	19.54 ^a^	15.64 ^ab^	12.95 ^b^	2.01	0.03
Lactase, U/g	316.86 ^a^	268.51 ^b^	259.26 ^b^	5.90	<0.01
Maltase, U/mg	177.49 ^a^	146.19 ^b^	132.75 ^b^	8.99	<0.01
Sucrase, U/mg	184.99 ^a^	163.38 ^ab^	143.78 ^b^	12.10	0.02
Ileum					
Amylase, U/g	99.78	49.93	34.29	29.47	0.12
Lipase, U/mg	14.24	13.27	11.83	0.98	0.10
Chymotrypsin, U/mg	10.41	9.71	8.56	0.98	0.22
Trypsin, U/mg	15.40	13.59	11.41	2.31	0.28
Lactase, U/g	565.69 ^a^	488.10 ^b^	423.99 ^c^	14.87	<0.01
Maltase, U/mg	129.43 ^a^	104.59 ^ab^	91.61 ^b^	13.17	0.05
Sucrase, U/mg	133.28	107.31	95.96	13.54	0.06

^1^ Values are the means of 6 observations. BR1 = Brown rice stored for 1 year; BR6 = Brown rice stored for 6 years; SEM = standard error of the mean. ^a, b^ Different superscript letters mean significantly different (*p* < 0.05).

**Table 9 animals-13-03381-t009:** Effects of stored brown rice on growth performance in the fully grown pigs (Exp. 4) ^1^.

Item	Phase	Control	BR1	BR6	SEM	*p* Value
BW, kg	Initial	47.85	48.55	48.43	0.38	0.19
	End Phase I	71.55 ^b^	74.25 ^a^	72.95 ^ab^	0.72	<0.01
	End Phase II	97.63	98.00	98.83	1.56	0.74
ADG, kg/d	Phase I	0.91	0.99	0.94	0.03	0.10
	Phase II	0.97	0.88	0.96	0.07	0.45
	Total	0.94	0.93	0.95	0.03	0.84
ADFI, kg/d	Phase I	2.61 ^b^	2.89 ^a^	2.94 ^a^	0.08	<0.01
	Phase II	3.05	3.10	3.20	0.08	0.23
	Total	2.90 ^a^	3.01 ^ab^	3.10 ^b^	0.07	0.03
FCR	Phase I	2.86	2.94	3.12	0.14	0.23
	Phase II	3.18	3.56	3.38	0.24	0.32
	Total	3.09	3.24	3.26	0.08	0.13

^1^ Values are the means of 6 observations. BR1 = Brown rice stored for 1 year; BR6 = Brown rice stored for 6 years; SEM = standard error of the mean; BW = body weight; ADG = average daily gain; ADFI = average daily feed intake; FCR = feed conversion ratio. ^a, b^ Different superscript letters mean significantly different (*p* < 0.05).

**Table 10 animals-13-03381-t010:** Effects of stored brown rice on carcass traits and meat quality of fully grown pigs (Exp. 4) ^1^.

Item	Control	BR1	BR6	SEM	*p* Value
Carcass traits					
Dressing percentage, %	65.67	69.20	70.02	1.67	0.06
Carcass straight length, cm	104.38	106.25	103	3.08	0.59
Backfat thickness, cm	2.39	1.98	2.47	0.26	0.19
Loin–eye area, cm^2^	46.95	54.46	49.81	8.11	0.66
Meat quality					
Drip loss, %	2.41	2.27	3.08	0.37	0.12
Shear force, N	36.97	40.88	37.85	4.99	0.72
pH45 min	6.39	6.54	6.18	0.21	0.26
pH24 h	5.92	5.84	5.81	0.16	0.81
L* (lightness)	42.28	42.35	43.60	1.20	0.50
a* (redness)	13.86	13.22	14.25	0.84	0.50
b* (yellowness)	2.02	2.40	1.85	0.47	0.52

^1^ Values are the means of 6 observations. BR1 = Brown rice stored for 1 year; BR6 = Brown rice stored for 6 years; SEM = standard error of the mean; BW = body weight; ADG = average daily gain; ADFI = average daily feed intake; FCR = feed conversion ratio.

**Table 11 animals-13-03381-t011:** Effects of stored brown rice on fatty acids profiles in the longissimus dorsi muscle of fully grown pigs (mg/g, of fresh tissue) (Exp. 4) ^1^.

Item	Control	BR1	BR6	SEM	*p* Value
Capric acid (C10: 0)	0.16	0.16	0.16	0.02	0.97
Lauric acid (C12: 0)	0.09	0.09	0.09	0.01	0.58
Myristic acid (C14: 0)	1.30	1.36	1.38	0.21	0.63
Palmitic acid (C16: 0)	21.62	22.15	23.03	8.36	0.98
Palmitoleic acid (C16: 1)	3.41	3.22	3.40	0.22	0.64
Heptadecanoic acid (C17: 0)	0.34	0.34	0.33	0.03	0.98
Stearic acid (C18: 0)	12.24	12.74	12.88	3.18	0.99
Oleic acid (C18: 1n-9c)	39.79	39.07	39.82	9.83	0.92
Linoleic acid (C18: 2n-6c)	7.72	7.98	7.98	1.14	0.84
Alpha-linolenic acid (C18: 3n-3)	0.37	0.39	0.36	0.03	0.87
Gama-linolenic acid (C18: 3n-6)	0.11	0.10	0.12	5.19	0.82
Icosanoic acid (C20: 0)	0.25	0.26	0.25	0.52	0.89
Eicosenoic acid (C20: 1)	0.74	0.72	0.75	0.05	0.93
Decosahedaenoic acid (C20: 2)	0.48	0.48	0.63	0.02	0.71
Dihomo-γ-linolenic (C20: 3n-6)	0.21	0.20	0.19	0.02	0.79
Arachidonic acid (C20: 4n-6)	1.53	1.45	1.69	0.06	0.60
Heneicosanoic acid (C21: 0)	0.34	0.33	0.35	0.20	0.87
n-6/n-3 PUFA	25.79	25.43	27.68	3.44	0.79
PUFA/SFA	0.33	0.33	0.34	0.11	0.99

^1^ Values are the means of 6 observations. BR1 = Brown rice stored for 1 year; BR6 = Brown rice stored for 6 years; SEM = standard error of the mean; PUFA = polyunsaturated fatty acid; SFA = saturated fatty acid.

## Data Availability

Data is contained within the article.

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
