# Peer review of "Evaluation of the Available Energy Value and Amino Acid Digestibility of Brown Rice Stored for 6 Years and Its Application in Pig Diets"

_animals, 2023, doi:10.3390/ani13213381_

Round 1

Reviewer 1 Report

Comments and Suggestions for Authors

The manuscript (animals-2636111) evaluated the nutritional quality of brown rice stored for 6 years and its application in pig diets. The study found that the digestible energy and metabolizable energy, as well as standardized ileal digestibility of amino acids, were not changed in brown rice stored under good conditions for 6 years. Stored brown rice also had no significant effects on growth performance, nutrient apparent total tract digestibility, serum biochemical indices, carcass traits, meat quality, and muscle fatty acids profiles of pigs, but reduced the activity of digestive enzymes in pig small intestine. The study concluded that brown rice stored under good conditions for 6 years can be used as a high-quality energy feed raw material in pig diet. I evaluate this manuscript as a major revision, as several points need to be carefully revised before the next resubmission as follows:

-  Line numbering is missing. Please, add to facilitate reviewing of the manuscript.

-    In The discussion section, the writing style should be formal from the third-person perspective. Do not use “we” or “our”. 

-    In the Abstract, the numbering should be removed from the results and the results should be presented as consecutive sentences.

-   “Rice is the most important grain crop and the annual output has reached more than 200 million tons in China.” this sentence needs to be supported by one or two recent citations.

-    “The paddy rice used in the present study were stored in brick concrete barns” Add more information about concrete barns such as their dimensions, capacity... etc

-   Is there any available information regarding the relative humidity range during the storage period?

-  Neutral detergent fiber (NDF) and acid detergent fiber (ADF) were determined using the ANKOM200 Fibre Analyzer” Specify If you added α amylase or not, and was their content expressed as exclusive or inclusive of residual ash?

-   “The fatty acid value was analyzed according to the procedures of GB/T 20569-2006 (Guidelines for evaluation of paddy storage character).” add to the references list.

-     “….to meet the nutrient requirements for growing pigs recommended by the National Research Council (NRC, 2012) (Table A2).” Missing in the references list. Also, the citations throughout the manuscript should be modified to comply with the Journal style.

-   “0.3% chromic oxide” Provide full details of chromic oxide, ie, formula, purity, company, city, country.

-   The identification of the 3 treatment diets should be mentioned in the M&M section.

-     In Experiment 1 and 2, were the piglets the same or different?

-  Please clarify the sex (male, female, or both) of the pigs used in all experiments.

-          In Experiment 4, Describe the pig pens.

-   Common analysis of amino acids cannot differentiate between asparagine and asparagic acid, as well as glutamine and glutamic acid, and report the sum of the respective pairs of amino acids (usually addressed to as “Asx” and “Glx”). Please check the analytical data and adapt the text accordingly.

-    How were blood samples collected and at which time of the experiment?

Comments on the Quality of English Language

-

Author Response

Thank you very much for taking the time to review this manuscript. Please find the detailed responses in the attachment and the corresponding revisions/corrections highlighted in the re-submitted files.

Reviewer 2 Report

Comments and Suggestions for Authors

The present paper is well written, clear and it is described the effect of different diets in pigs. In my opinion it would be interesting to evaluate if these diets have some effects on the intestinal anatomy/physiology performing histological stainings for example

Author Response

Thank you very much for taking the time to review this manuscript. We have made some modifications to the manuscript, please find the corresponding revisions/corrections highlighted in the re-submitted files.

Reviewer 3 Report

Comments and Suggestions for Authors

General Comments:

This paper presents valuable research on the effects of stored brown rice on animal nutrition and performance. Addressing the above points would further enhance the quality and clarity of the paper.

In Introduction section: 

It would be helpful to provide some statistics or data related to the increasing contradiction between corn supply and demand to support the claim made in the introduction.

Moreover, is storage rice increasing in China? The authors should add the information in the manuscript. Furthermore, why the authors choose storage for "6 years".

In Materials and Methods section:

It would be helpful to include more information about the specific conditions in which the brown rice was stored for 1 or 6 years, as this could be a critical factor affecting the results.

Clarify the how to make the experimental feed. Explain how this choice might affect the results.

Did the authors fed crused rice? Made pellet? If the authors fed uncrushed rice, the digestibility is high.Moreover, The tables about feed should not be in the appendix.

Furthermore, how to analyze the amino acids? Using HPLC? GC-MS?

In Discussion section:

The discussion provides a good interpretation of the results. However, it could benefit from more comparisons with previous research in this field. Are the findings consistent with or different from other studies? This would help provide context and validate the results.

Discuss the practical implications of the findings. How can the results be applied in the field of animal husbandry or livestock production?

Consider discussing any limitations of the study, such as potential sources of error or bias, and how these limitations might have affected the results.

Author Response

(The authors gave the same response as above.)

Round 2

Reviewer 1 Report

Comments and Suggestions for Authors

The authors adequately responded to all comments and performed all required modifications.

Reviewer 3 Report

Comments and Suggestions for Authors

The authors addressed successfully my comments and the manuscript can be published in its current form.